Water lettuce (Pistia stratiotes L.) increases biogas effluent pollutant removal efficacy and proves a positive substrate for renewable energy production

http://orcid.org/0000-0003-4717-6575 Nguyen Vo Chau Ngan 1 nvcngan@ctu.edu.vn
http://orcid.org/0000-0001-6582-9061 Huynh Van Thao 2 3
Nguyen Cong Thuan 2
Kim Lavane 4 klavane@ctu.edu.vn
Pham Dan Van 5
1 Department of Water Resources, Can Tho University , Can Tho City , Vietnam
2 Department of Environmental Sciences, Can Tho University , Can Tho City , Vietnam
3 United Graduate School of Agricultural Science, Tokyo University of Agriculture and Technology , Tokyo , Japan
4 Department of Environmental Engineering, Can Tho University , Can Tho City , Vietnam
5 Center for Technology Development and Agricultural Extension, Vietnam Academy of Agricultural Sciences , Ha Noi , Vietnam
Phitsuwan Paripok
Electronic publication date: 2023 Aug 22
Publication date: 2023
Volume: 11
Electronic Location ID: e15879
Received 2023 Feb 9; Accepted 2023 Jul 19
Copyright: © 2023 Nguyen Vo Chau et al.
Copyright year: 2023
Copyright holder: Nguyen Vo Chau et al.
License: This is an open access article distributed under the terms of the Creative Commons Attribution License, which permits unrestricted use, distribution, reproduction and adaptation in any medium and for any purpose provided that it is properly attributed. For attribution, the original author(s), title, publication source (PeerJ) and either DOI or URL of the article must be cited.
License URL: https://creativecommons.org/licenses/by/4.0/

Keywords: Biodegradation, Husbandry waste, Phytoremediation, Wastewater treatment, Water lettuce

Funding: The authors received no funding for this work.

==============================
Background

Aquatic plants play a crucial role in nature-based wastewater treatment and provide a promising substrate for renewable energy production using anaerobic digestion (AD) technology. This study aimed to examine the contaminant removal from AD effluent by water lettuce (WL) and produce biogas from WL biomass co-digested with pig dung (PD) in a farm-scale biogas digester.

Methods

The first experiment used styrofoam boxes containing husbandry AD effluent. WLs were initially arranged in 50%, 25%, 12.5%, and 0% surface coverage. Each treatment was conducted in five replicates under natural conditions. In the second experiment, WL biomass was co-digested with PD into an existing anaerobic digester to examine biogas production on a farm scale.

Results

Over 30 days, the treatment efficiency of TSS, BOD5, COD, TKN, and TP in the effluent was 93.75–97.66%, 76.63–82.56%, 76.78–82.89%, 61.75–63.75%, and 89.00–89.57%, respectively. Higher WL coverage increased the pollutant elimination potential. The WL biomass doubled after 12 days for all treatments. In the farm-scale biogas production, the biogas yield varied between 190.6 and 292.9 L kg VSadded−1. The methane content reached over 54%.

Conclusions

WL removed AD effluent nutrients effectively through a phytoremediation system and generated significant biomass for renewable energy production in a farm-scale model.

Introduction

Agricultural production is the main economic activity in the Vietnamese Mekong Delta (VMD), of which livestock production constitutes more than 20% (General Statistics Office of Vietnam, 2021). Within this sector, pig livestock dominates in most local areas. It is estimated that the VMD has 2.08 million pigs (General Statistics Office of Vietnam, 2022). Mostly, pig farms temporarily raise livestock on a small scale (<10 pigs) without suitable waste treatment measures (Nam et al., 2021). Therefore, waste management from pig raising is a significant challenge owing to the considerable amount of pig excrement directly released to the external environment (Ngan, 2012; Nam et al., 2022). Environmental issues have emerged seriously in rural areas, where local communities use surface water sources from rivers/canals as the primary water supply source for agriculture production and domestic utilization (Roubík et al., 2018; Kamyab et al., 2022). Consequently, sustainable livestock production requires responsible waste management to minimize negative influences on the surrounding environment and ecosystems.

Anaerobic digestion (AD) is an effective technology for treating biodegradable waste and producing eco-friendly energy (Markphan et al., 2020; Ye et al., 2013; Szaja et al., 2020; Wang et al., 2022). In the 1990s, this technology was introduced in the VMD to treat animal waste and recapture biogas for cooking and heating (Ngan, Hieu & Nam, 2012). However, the scale of raising livestock of the VMD’s households is modest, leading to a shortage of feedstock substrate for producing biogas (Nam et al., 2022). Accordingly, substitute or additional feedstocks, such as rice straw, water hyacinth, and potential biomass, are encouraged for use as co-substrates for biogas digesters to achieve higher biogas yields (Nam, Van Cong & Van Thao, 2023). The co-digestion of animal waste and biomass provides a more flexible digestion process in small livestock household digesters. This approach potentially enhances biogas production by adjusting the carbon/nitrogen (C/N) ratio to a more favorable range for anaerobic biodegradation regression (Nam et al., 2021; Yadvika et al., 2004).

Although biogas production is offered not only for livestock waste treatment but also for renewable energy production, the AD process shows limitations in waste treatment efficiency and outlet effluent quality when solely substrate is applied. The AD effluent quality regularly does not meet the national technical regulations on husbandry wastewater (QCVN 62-MT:2016/BTNMT) before it is released into the exterior environment (The Ministry of Natural Resources and Environment of Vietnam, 2016). It is commonly accepted that treatment efficacy from AD is about 30% of the organic matter, while the rest remains as digestate and sludge (Gurung, 1997). Consequently, AD effluent contains much more indigestible matter and high concentrations of identified nutrients, such as nitrogen, phosphorous, and potassium (Ngan, 2012; Abe et al., 2016; Wang et al., 2018; Kalaimurugan et al., 2022). As such, reutilizing nutrients from wastewater or AD effluent to grow aquatic plants is a promising option for bioremediation because it is cheap and eco-friendly (Nor et al., 2023).

In the rural areas of the VMD, households typically own a large plot of land to develop a traditional farming system known as VAC (V—garden, A—pond, and C—livestock cage) or VACB (V—garden, A—pond, C—livestock cage, and B—biogas plant) (Ngan, 2011). The pond is commonly used as a buffer unit for receiving AD effluent before discharging it into the external environment. Ordinarily, several types of aquatic plants, such as water lettuce (Pistia stratiotes), water hyacinth (Eichhornia crassipes), and duckweed (Lemna minor), are grown on the pond to eliminate onsite AD effluent nutrients. These plants are identified as invasive plants that are grown popularly in the tropical climate of the VMD (Coelho, Deboni & Lopes, 2005; Šajna et al., 2007). In addition, these plants are known for their efficient nutrient-absorbing ability, metal and toxic elimination, and biogenic element accumulation (Lu et al., 2010; Brix, 1991). Aquatic plant biomass could be valuable for promoting renewable energy production through AD technology (Barua & Kalamdhad, 2019; Nam et al., 2021).

Although water hyacinth is a good candidate for nutrient removal and biomass harvest, its high content of hemicellulose (30.8–48.0%) slows biogas production, resulting in overloading in a long-term operation (Nam, Van Cong & Van Thao, 2023; Lin et al., 2015; Bote, Naik & Jagadeeshgouda, 2020). The possibility of duckweed biomass production is slight, albeit with high biomethanation potential, resulting in an inadequate substrate for household-scale biogas production. In contrast, water lettuce (WL) has demonstrated its effectiveness for domestic wastewater contaminant removal of BOD, NH3, TN, and TP by 83.5–97.53% in a 5-day treatment (Gaballah et al., 2019). The rapid elimination of nutrients in wastewater can reduce the risk of water body eutrophication. Moreover, WL has great biomass production potential due to its fast-growing nature. WL’s biomass production is about 2.4 tons of dry biomass per hectare per year, indicating stable substrate-providing feasibility for biogas production (Sutaryo et al., 2022). This plant can also grow in various environments, including clean water and wastewater, signifying its adaptability for AD effluent nutrients in varying amounts (Whangchai et al., 2021). WL’s chemical composition is 34.4% cellulose, 11.6% lignin, and 26.3% hemicellulose, which is promising for renewable energy production from biodegradation compared to other agricultural terrestrial plants (Güngören Madenoğlu et al., 2019). Previous studies have demonstrated the concordance of biogas production from WL exclusively and from co-digestion with cow dung and waste sludge (Cong et al., 2022; Sutaryo et al., 2022; Güngören Madenoğlu et al., 2019). Biogas yields of WL range from 533–707 L kg VS–1 (Güngören Madenoğlu et al., 2019), while the potential of biogas production from water hyacinth and duckweed varies from 102 to 478 L kg VSadded−1 (Nam et al., 2017; Patil et al., 2014; Gunnarsson & Petersen, 2007) and from 250 to 390 L kg VSadded−1, respectively (Tonon et al., 2017). As such, WL is promising for contaminant removal, as well as its availability and sustainable use for biogas production at a farm scale. Although WL’s potential has been identified, the effectiveness of removing contaminants in AD effluent and biogas production from co-digestion of WL with pig manure at the household scale has remained unclear. This study, therefore, aimed to (i) examine the contaminant removal of WL applied to AD effluent and (ii) investigate the biogas production potential of WL co-digested with pig manure in a farm-scale biogas digester.

Materials and Methods

Study site

All the experiments were implemented at Mr. Liem Le Hoang’s farm in Thanh My village (9°59′24.44″, 105°47′43.57″), Cai Rang district, Can Tho city, Vietnam. The household had raised pigs for a long time and owned a 2-year-old HDPE (high density polyethylene) biogas tube plant of 7.64 m3 (Fig. 1), but only one pig of 40 kg was being raised in the pigsty at the study implementation time. The field experiments were approved by the College of Environment and Nature Resources, Can Tho University (ID 101/KMT).

Figure 1 The existing polyethylene biogas digester.

Experimental design

Experiment 1: water lettuce phytoremediation performance

The WL was collected from the adjacent pond to the experiment site. The plants chosen varied from 4–6 cm in length, and each plant had from five to seven leaves at the initiation. We removed all the bad parts from the plants and rinsed them with tap water to eliminate any debris of other macrophytes. Then, the WL was placed into a plastic barrel filled with husbandry anaerobic digester effluent (HADE) for 1 week, allowing it to adapt to its environmental conditions.

The experiment was designed in a batch system using styrofoam boxes (60 cm long × 45 cm wide × 41 cm high). Each box contained 30 L of HADE, which was not added to during the experiment. The WL was placed into boxes for different treatments by the plants arranged in 50%, 25%, and 12.5% of the box’s surface area. In parallel, a control treatment (0%) was set up without WL. Each treatment was replicated in five boxes. Accordingly, a total of 20 boxes were prepared. All the experimental boxes were randomly arranged on a scaffold under natural conditions. A plastic film was set up to cover the experiment boxes if it rained. The experiment ended in 30 days. The pollution elimination percentage was calculated as shown in Eq. (1):

(1) Removalefficiency(%)=Ci−CeCi×100

in which Ci is the initial concentration of pollutants and Ce is the final concentration of pollutants remaining in the effluent.

The HADE was collected from the existing PE biogas plant at the experiment site and installed in the boxes without any dilution. The HADE was examined before the initial experiment and after the phytoremediation tests at 6, 12, 18, 24, and 30 days. The water samples were collected as mixed samples from each treatment to achieve a pooled sample. The physicochemical parameters of pH, TSS, BOD5, COD, total Kjeldahl nitrogen (TKN), and total coliforms were chosen as required by the QCVN 62-MT:2016/BTNMT National Technical Regulation on Livestock Effluent. The collected HADE samples were analyzed according to the Standard Methods for the Examination of Water and Wastewater (APHA, 1998).

The growing ability of the WL was recorded before and after the phytoremediation process every 5 days for the 30 days of the experiment. The plants’ fresh weight was determined using a digital balance to weigh all the plants within each box. The root length and leaf length were measured by randomly choosing three plants from each box and recording the average value of the three maximum values of the root and leaf (Fig. 2).

Figure 2 Water lettuce biomass measurement.

Experiment 2: anaerobic digestion performance

WL from the nearby pond was collected and fed into the existing PE biogas plant as Supplemental Material. After each collection time, the WL was dried within a day in shady conditions to avoid nutrient loss, and it was then fed into the biogas digester. The quantity of WL co-fed every 2 days into the HDPE biogas plant was 10 kg wet weight within the first 10 days, 40 kg wet weight in the next 30 days, and 24 kg wet weight in the last 20 days. At the start, the volatile solids (VS) content of the WL was determined by drying a sample of the biomass at 105 °C for 24 h and then heating it at 550 °C for 2 h. The VS value was then calculated based on the weight of the dried sample.

The daily produced biogas was recorded onsite by a VIGADO G1.6 gas meter (PDM 095-2008; Vietnam). For the gas components, biogas samples were collected in an aluminum bag after feeding the WL every 4 days for 60 days. The gas samples were transported to the biogas laboratory at Can Tho University and analyzed using a portable biogas analyzer (Biogas Pro 5000; Landtec, Dexter, MI, USA) for the biogas components, including CH4, CO2, and other compositions (Fig. 3).

Figure 3 Measurement of biogas volume and biogas composition.

Data processing

The data were plotted using the RStudio software package (RStudio Team, 2023). The Pearson correlation coefficient was employed to assess the correlation between the HADE pollutant variables (TSS, BOD5, COD, TKN, and TP) and the WL biomass (leaf length, root length, and fresh weight biomass). The correlation significance is indicated as follows: an asterisk (*) represents 0.05 < P < 0.01, two asterisks (**) represent 0.01 < P < 0.001, and three asterisks (***) represent P < 0.001.

Results and discussion

Initial HADE characteristics

The HADE collected initially and used for the phytoremediation experiment had a high organic concentration compared to the National Technical Regulation on Livestock Effluent (Table 1). Compared to a previous study on the effluent of similar PE biogas plants operated in the VMD (Ngan, 2012), the tested values were lower, as only one pig was being raised at the time of the study. However, HADE is well suited for the WL phytoremediation process. Thus, the collected HADE was initially applied to set up the experiment without dilution. Within the study period, the daily ambient temperatures (recorded randomly between 08:00 and 16:00) at the experiment site ranged from 27.3 °C to 30.7 °C (Fig. 4). The variation was appropriate for the optimum temperature (22–30 °C) for growing WL (Pettet & Pettet, 1970). The water levels in the experimental boxes were also recorded to ensure the growing conditions for the WL. At the beginning, all the experimental boxes were filled with a water level of 21 cm. At the end of the experiment time, the water levels were 14.54, 15.54, 15.52, and 16.40 cm for the WL surface cover of 50%, 25%, 12.5%, and 0%, respectively. Thus, the water loss was estimated at 0.93, 0.82, 0.81, and 0.72 cm day−1 under 50%, 25%, 12.5%, and 0% WL surface cover (Fig. 5).

Table 1 The physiochemical properties of husbandry anaerobic digester effluent.

Parameter	Unit	Concentration	Reference	Standard↨	
pH	–	7.33	6.5–7.4†	5.5–9.0	
TSS	mg L−1	128	70–12,200†	50	
BOD5	mg L−1	86	360–1,125.4‡	40	
COD	mg L−1	149	84–5,864†	100	
TKN	mg L−1	40	16.8–548.8†	50	
TP	mg L−1	14	5.5–122.3†	NA	
Total coliform	MPN 100 mL−1	7.5 × 105	4.6 × 104–9.3 × 107‡	3,000	
Notes:

† Final report on environmental and climate sound adaptation of biogas plant in Vietnam (UKAVita); N. V. C Ngan, N. X. Hoang, 2020, unpublished data.

‡ Ngan (2012).

↕ QCVN 62-MT:2016/BTNMT (referred to A column).

Figure 4 Ambient temperature during experiment.

Figure 5 The variation of husbandry anaerobic digester effluent in treatments over times.

Phytoremediation process experiment

Pollution removal

Figure 6A shows the phytoremediation performance of the HADE using WL every 6 days for 30 days. The pH levels slightly decreased during the phytoremediation process for all treatments. Although the treatment with 12.5% surface cover showed the largest change in pH values between days 12 and 18, the difference in pH across all treatments was negligible.

Figure 6 (A–F) Husbandry anaerobic digester effluent polluted parameters removal by times.

The concentration of contaminants (TSS, BOD5, COD, TKN, and TP) rapidly reduced in the first 6 days of running the experiment (Figs. 6B–6F). There was a slight decrease for the rest of the period. The higher coverages of WL displayed better treatment efficacy of HADE. All the WL treatments demonstrated a higher potential for pollutant elimination than the control treatment (0%) (Fig. 6).

Figure 7 shows the pollutant treatment efficiency of WL in the effluent from the digester. In the first 6 days, there was a sharp decrease in the treatment efficiencies of the WL, but then a slight decrease between days 6 and 30 was shown. The TSS removal efficiency in the first 6 days of TSS was highest at 80.47–94.53% (Fig. 7A), followed by TP (68.36–82.86%, Fig. 7E), BOD5 (44.19–59.30%, Fig. 7B), COD (44.3–59.73%, Fig. 7C), and TKN (38.25–46.75%, Fig. 7D). After 30 days, the reduction efficiency of TSS and TP did not continuously increase much compared to the first 6 days, while the treatment efficiencies of BOD5, COD, and TKN rose much higher than during the first 6 days (Fig. 7).

Figure 7 (A–E) Treatment efficiency of husbandry anaerobic digester effluent.

Recent studies have confirmed high treatment efficiency on BOD5 (83.3–92.8%) and COD (79.2–85.9%) when the phytotreatment of polluted river water, palm oil mill wastewater, and sewage wastewater using macrophytes was applied (Shahid et al., 2019; Wei, 2019; Schwantes et al., 2019). However, other research has reported lower removal efficiencies of phytotreatment. Jyotsna, Bhasin & Punit (2015) confirmed removal efficiencies of 44.8% of TSS, 62.4% of BOD5, and 63.8% of COD for pulp and paper mill effluent after 30 days by lesser duckweed. Tang et al. (2009) recorded reduction efficiencies of 33.2% of COD and 21.8% of TKN for wetland constructed to treat eutrophic river water. However, the organic concentrations in the pulp and paper mill effluent and eutrophic river water were much higher than in this study. Treatment effectiveness can vary depending on various factors, such as the initial pollutant levels, the duration of treatment, and the specific conditions of the treatment environment.

In our study, the removal efficiency of COD was slightly lower than that of BOD5, which is similar to the study by Hamzah, Yusof & Alimon (2016) of phytoremediation testing on palm oil mill final discharge wastewater (84.7% of BOD5 and 22.3% of COD). In fact, recalcitrant compounds, such as high humic acid in pig dung (PD), are known to cause low treatment efficiency of COD of the phytoremediation (Norulaini, Ahmad Zuhairi & Muhamad Omar, 2001; Zhang et al., 2012).

High removal efficiencies of over 89% of TP were recorded for all the WL treatment coverages after 30 days. The utility of phosphorus in aquatic plants is essential in various physicochemical and biological processes. In addition, values from 61.8% to 65.0% were the reduction efficiencies of TKN after 30 days of phytoremediation. In a similar approach using swine wastewater, Sudiarto, Renggaman & Choi (2019) documented that TKN and TP removals within 21 days were 71.4% and 32.2%, respectively. Similarly, Parwin & Paul (2019) reported the highest removal efficiencies of 94.4% of TKN and 98.1% of TP using water hyacinth to treat kitchen wastewater within 4 weeks. In contrast, the lowest reduction efficiencies of BOD5, COD, and TKN using water caltrop to treat municipal wastewater have been recorded as 18.3%, 14.2%, and 27.2%, respectively (Kumar & Chopra, 2018).

Water lettuce biomass

Previous findings disclosed that the biomass of WL increased up to the first 10 days and then diminished as the basal leaves decayed (Fonkou et al., 2002) or due to their maximum uptake limit of nutrients (Hamzah, Yusof & Alimon, 2016). Our study showed that the biomass gradually increased through the experiment period (Fig. 8). The difference compared to the previous report was due to the required space for the growth of WL. Based on the produced WL, the doubling time for the wet-weight biomass from the treatments of 50%, 25%, and 12.5% surface coverage were estimated at 12.32, 12.11, and 12.28 days, respectively (Table 2). The rapid increase in WL’s biomass indicates its potential for eliminating pollutants and providing valuable material for biogas production.

Figure 8 Water lettuce biomass change among treatments.

Table 2 Doubling time of water lettuce among treatments.

Treatments	Initial day	Final day	Doubling time (days)	
50%	70.8	536.4	12.32	
25%	39.2	307.6	12.11	
12.5%	19.2	146.4	12.28	
Average	–	–	12.24	

Figure 9A shows that the root lengths of the WL increased over time. Root lengths were recorded from 6.66 to 8.08 cm (an increase of 21.3%), 6.52 to 8.06 cm (an increase of 23.6%), and 6.16 to 8.04 cm (an increase of 30.5%) after 30 cultivated days for the treatments of 50%, 25%, and 12.5% surface coverage, respectively. The maximum leaf length of growth recorded in the treatment of a surface cover of 12.5% WL show that with fewer bodies (Fig. 9B). The nutrients were absorbed and formed up to the root system instead of the leaves. During the first 15 days, the length of the WL’s leaves increased in all treatments but then decreased as a result of the decay of the lower leaves and the emergence of new leaves. Consequently, a high amount of biomass was recorded as the WL continuously grew in the experimental containers.

Figure 9 The (A) root length and (B) leaves length of water lettuce.

Figure 10 depicts the Pearson correlation of the pollutant parameters in the HADE among the treatments, as indicated by the color and size of the circles. The magnitude of the correlation is reflected by these visual cues. A strong positive relationship was found between pollutant concentrations, with r values ranging from 0.8 to 1.0. The BOD5 and COD displayed the highest correlation. The Pearson correlation revealed adequate interdependence between the pollutant composition and the prospect of WL to effectively eliminate pollutant levels in the HADE. Similarly, leaf length and root length showed a moderate correlation (r = 0.53), while root length and fresh biomass exhibited a high correlation (r = 0.85) (Fig. 11). No relationship was found between leaf length and fresh biomass. As such, the root length played a crucial role in increasing the WL’s biomass. Macrophyte roots are vital for reducing contaminants in the aboveground part of their bodies. Moreover, macrophytes also secrete biopolymer from their roots, which assist in flocculation (Sharma, Singh & Manchanda, 2015). Non-settling and colloidal particles are also removed, at least partially, by bacterial growth, which results in the removal of some colloidal solids and the microbial decay of other organic pollutants.

Figure 10 Pearson correlation of pollutant parameters.

Note: Pearson correlation significant indicates as follows: *, 0.05 < P < 0.01; **, 0.01 < P < 0.001; and ***, P < 0.001.

Figure 11 Pearson correlation of water lettuce biomass.

Note: Pearson correlation significant indicates as follows: *, 0.05 < P < 0.01; **, 0.01 < P < 0.001; and ***, P < 0.001.

Biogas production from water lettuce biomass

The biogas produced from the HDPE digester is shown in Fig. 12. The analyzed WL result showed that the VS value was 4.9%, which was higher than the VS value reported by Abbasi & Nipaney (1991). For a co-substrate of WL and PD, feeding the material to the HDPE digester every 2 days is recommended (Ngan et al., 2018). In the current study, 10 kg of WL (approximately 0.49 kg of VS) was fed every 2 days for the first 10 days, allowing microorganisms to adjust slowly to the new feed. Afterwards, 40 kg of WL (about 1.96 kg of VS) was fed every 2 days for the next 30 days, and 24 kg of WL (equivalent to 1.18 kg of VS) was fed each time during the final 20 days. The loading rates of WL were low compared to those suggested by Eder & Schulz (2007) for the AD process, but PD was also fed into the biogas plant.

Figure 12 Biogas production from polyethylene digester.

On the first day of the WL feeding, 365 L day−1 of biogas production was recorded, which was transformed solely from the PD. On day 2, the produced biogas was slightly reduced (321 L day−1), announcing a small “shock” within the fermentation process. From days 3 to 10, biogas production increased and ranged from 266 to 597 L day−1 (411.7 ± 104.4 L day−1 on average). The biogas production volume increase due to the WL addition was estimated to be 46.7 L day−1. This amount is equivalent to a biogas yield of 190.6 L kg VSadded−1.

During the second period, biogas production decreased after increasing the feeding amount, as the microorganisms required some time to adjust to the large feedstock. The results show that a large variation in biogas production was recorded from 234 (day 14) to 1,280 L day−1 (day 32). Several peaks in biogas production appeared during the period. The average volume of biogas produced in this period was 676.2 ± 293.4 L day−1. One pig was being raised in the pigsty at that time. We assumed that the amount of PD increased by 5% compared to the first period due to the pig’s increasing weight. It is estimated that an amount [365 + (365 × 5%)] = 383.3 L day−1 of biogas was produced solely from the PD in the second period. Therefore, the biogas production volume increased due to the WL contribution by around 292.9 L day−1. As such, the biogas yields for the period were estimated at 292.9 L kg VSadded−1.

During the third period, biogas production gradually decreased compared to the second period. The maximum and minimum biogas productions were recorded on days 46 (1,039 L day−1) and 57 (260 L day−1). The average biogas production in this period was 518.6 ± 224.1 L day−1. If the amount of PD increased by 10% due to the growth of the pig, it is estimated that the biogas produced solely from the PD would amount to 401.5 L day−1 in the second period, calculated as [365 + (365 × 10%)] = 401.5 L day−1. Consequently, the average biogas volume produced from the WL was estimated to be approximately 117.1 L day−1. This resulted in an estimated biogas yield of 198.5 L kg VSadded−1.

The biogas yields increased gradually as the feeding material from the WL increased, demonstrating WL’s potential for renewable energy production. However, changes in the feeding amount can result in temporary disruptions, as microbes need time to break down the organic matter and nutrients in WL and convert them into biogas. It is worth noting that only a portion of the WL fed into the digester during the first 10 days was converted into biogas. In the next 30 days, the remaining WL from the previous period combined with the new feeding was digested, increasing biogas production in the second and third periods.

As shown in Fig. 12, the cumulative biogas production exhibits sigmoid growth curves, as described above. In the first phase, the curve showed a slow increase (lag phase) at the start of the co-feeding with WL. The second and third periods showed a rapid increase that approached exponential growth (log phase) due to the significant amount of WL added to the digester.

In this study, the biogas yields from WL fermentation ranged from 190.6 to 292.9 L.kg VS−1, comparable to previous reports involving the co-digestion of WL and cow manure and digestion of solely WL substrate (Phuong et al., 2015; Cong et al., 2022). It should be mentioned that these previous experiments were implemented in a lab-scale batch anaerobic digester with strict control over operating parameters, while this study was carried out in a farm-scale digester. Despite the differences in scale, this study demonstrated that WL is a promising material for biogas production.

Biogas composition

Figure 13 shows the change in biogas composition over time. The biogas composition recorded on the first day was 55.3% CH4 and 30.5% CO2. During the experiment, the CH4 content varied from 54.1% to 59.9% and the CO2 content varied from 21.9% to 31.7%. Nam et al. (2015) reported 47.7–56.6% and 46.5–54.4% methane content for onsite experiments of co-digestion of rice straw and water hyacinth, respectively, with PD using PE biogas plants in the VMD. Another study on co-digestion of cow dung and WL found CH4 content from 50.6% to 54.8% after 2 weeks of fermentation (Phuong et al., 2015), while Cong et al. (2022) discovered that solely digesting WL produced the highest concentration of methane by 62.2% on day 35. This study noted a high methane content from the start of the experiment, as the HDPE plant had been in operation for 2 years and was being fed with PD. The biogas could be applied to energy consumption due to its high methane content, particularly for household cooking in the VMD.

Figure 13 Biogas composition from testing polyethylene digester.

Feasibility of farm-scale biogas production and AD effluent removal systems

Biogas production can be considered a renewable energy source in most localities. Unlike other renewable energy sources, such as solar and wind power, biogas energy sources are used directly by households. Accordingly, biogas plants are more appropriate for decentralized energy production areas in the VMD. It is notable that bio-gasification from AD depends on several factors, such as carbon/nitrogen (C/N), hemicellulose content, pH, and the buffering capacity of the substrate (Güngören Madenoğlu et al., 2019). In farm-scale experimentation, WL showed great biogas production potential for co-digestion with PD. CH4 concentration was achieved by more than 50% after 2 weeks, implying conformity for household cooking and heating (Ngan et al., 2020).

Under these circumstances, WL source availability is the main challenge in expanding the system for households. In the VMD, many household farms have sufficient land area to install biogas, as well as a pond for growing WL combined with aquacultural activities. WL can be grown on a pond with a nutrient source provided by the livestock biogas plant. The contaminant-eliminating productivity of WL was reliable throughout the current study. Nutrient removal by WL involves not only eutrophication reduction of surrounding water bodies, but also significant biomass production for renewable energy generation. WL biomass can be used to feed the biogas plant directly or applied as simple bio-pretreatment technologies (soaking in biogas effluent or anoxic mud for a 5-day period) to enhance the biogas yield, as suggested by Nam et al. (2021) and Nam, Van Cong & Van Thao (2023). The co-digestion of WL and PD could solve common issues concerning insufficient animal waste sources for biogas production, because many farms in various localities typically have a smaller standing stock of animals (permanently or temporarily) (Nam et al., 2017, 2021). Our study encourages the co-digestion of WL with PD to enhance renewable energy production. Notably, the use of renewable energy produced from AD allows for energy independence in the context of high energy consumption demand. Moreover, the application of household biogas plants to produce low-cost energy increases household economic efficacy owing to saving electrical use, liquefied petroleum gas (LPG) and firewood, while reducing greenhouse gas emissions from agricultural activities (Jan, Truc & Nam, 2018).

Conclusions

This study applied WL as a post-treatment method to AD effluent and achieved impressive removal efficiencies of pollutants after 6 days, including 84.4–94.5% TSS, 80.0–82.9% TP, 50.3–59.7% COD, 50.0–59.3% BOD5, and 38.3–46.8% TKN. Over 30 days, the treatment efficiencies of TSS, BOD5, COD, TKN, and TP in the effluent were 93.8–97.7%, 76.6– 82.6%, 76.8–82.9%, 61.8–63.8%, and 89.0–89.6%, respectively. Based on the wet weight biomass, the doubling times for WL grown on surface coverages of 50%, 25%, and 12.5% were 12.32, 12.11, and 12.28 days, respectively. Thus, WL could be used as a supplementary substrate for biogas production. The biogas yield was recorded from 190.6 to 292.9 L kg VSadded−1 when converted solely from WL. The CH4 content was verified from 54.1% to 59.9%, making it suitable for biogas consumption. In general, WL could be applied in the post-treatment stage for AD and as an additional substrate for biogas plants to produce renewable energy. Further research should explore the potential of using WL as a substitute substrate for livestock waste to generate energy.

Supplemental Information

Supplemental Information 1 Raw data.

Click here for additional data file.

We thank Nguyen Hoang Phuong at Can Tho University for assisting with the experiment activities.

Additional Information and Declarations

Competing Interests

Author Contributions

Field Study Permissions

Data Availability

The authors declare that they have no competing interests.

Ngan Nguyen Vo Chau conceived and designed the experiments, performed the experiments, analyzed the data, authored or reviewed drafts of the article, and approved the final draft.

Thao Huynh Van conceived and designed the experiments, prepared figures and/or tables, writing original draft, and approved the final draft.

Thuan Nguyen Cong performed the experiments, analyzed the data, authored or reviewed drafts of the article, and approved the final draft.

Lavane Kim conceived and designed the experiments, performed the experiments, authored or reviewed drafts of the article, and approved the final draft.

Dan Van Pham analyzed the data, prepared figures and/or tables, authored or reviewed drafts of the article, and approved the final draft.

The following information was supplied relating to field study approvals (i.e., approving body and any reference numbers):

Field experiments were approved by the College of Environment and Nature Resources—Can Tho University (ID code 101/KMT).

The following information was supplied regarding data availability:

Raw data is available in the Supplemental Files.

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
