# Peer review of "Water lettuce (Pistia stratiotes L.) increases biogas effluent pollutant removal efficacy and proves a positive substrate for renewable energy production"

_PeerJ, doi:10.7717/peerj.15879_

## Round 0.1 · original submission · Major Revisions

- Please indicate whether the water lettuce is promising regarding production scale, availability, and sustainability use as feedstock for methane production.

- Improve the Introduction and provide rationales for the research, particularly the contribution of water lettuce to biogas production compared to other plants.

- English should be improved.

·

Basic reporting

The work is average but may be improved by the inclusion of the following suggestions. The manuscript can be a fair contribution if properly revised. The quality of science is mediocre. As it is not very readable, most of the findings could be shown rather as Tables instead of text.
The writing and grammar of the manuscript should be carefully checked to resolve some existing errors and provide smooth text.
The aim and its application have not been clearly described. What are the environmental benefits of using water lettuce for biogas production and pollutant removal?

Experimental design

Acceptable

Validity of the findings

How does the use of water lettuce as a substrate for biogas production compare to other renewable energy sources, such as solar or wind power?
How can water lettuce be integrated into wastewater treatment systems to improve energy efficiency and sustainability? How can we optimize the growth and production of water lettuce to increase its efficacy as a renewable energy substrate and pollutant remover?
What are the economic implications of using water lettuce for biogas production, and how do they compare to traditional fossil fuel sources?

Additional comments

In the Introduction section, please outline the main aim, objectives, and research questions clearly and articulate the research questions to the significance of the work. The review of the literature needs more updating with works to have a clear and concise state-of-the-art analysis. You may see and cite these articles: https://www.mdpi.com/2071-1050/14/4/2142; https://link.springer.com/article/10.1007/s00449-022-02749-1; https://www.sciencedirect.com/science/article/abs/pii/S0045653522029642; What is water lettuce, and how does it contribute to biogas effluent pollutant removal? What factors affect the growth and production of water lettuce as a renewable energy substrate?

·

Basic reporting

In general, this manuscript is well-written, concise, and presents relevant information based on theory. While the methods used are generally appropriate, there are some details that require clarification. Overall, the study's results are clear and compelling, with only a few minor exceptions. The authors make a systematic contribution to the research literature in this area of investigation.

Experimental design

Used appropriate methods

Validity of the findings

accepted

Additional comments

Specific comments will follow,
1.The authors need to incorporate more content in introduction section regarding AD effluent characterization with suitable reference

2. Water hyacinth has good pollution removal potential. Why is water lettuce used for this study? Why not water hyacinth?

3. Is any microbe (living in the root of lettuce) involved in the treatment?

4. What is the reason for higher production of methane from water lettuce? Add the scientific reason in biogas composition section with appropriate scientific evidence.

---

## Round 0.2 · accepted · Accept

Your manuscript is much improved and it can be accepted for publication.

·

Basic reporting

The paper is well –written/clear and unambiguous. As per previous reviewer comments, required modification has done. Structure conforms to PeerJ standards, discipline norm.

Original research

Experimental design

Appropriate methods are used

Validity of the findings

Obtained data are valid

Additional comments

This original research paper may attain the required quality/ standards to publish in Peer J journal.